# Tuning the Surface Functionality of Fe_3_O_4_ for Sensitive and Selective Detection of Heavy Metal Ions

**DOI:** 10.3390/s22228895

**Published:** 2022-11-17

**Authors:** Manjunatha Kumara K. S., D. H. Nagaraju, Zhoveta Yhobu, Nayan Kumar H. N., Srinivasa Budagumpi, Shubhankar Kumar Bose, Shivakumar P., Venkata Narayana Palakollu

**Affiliations:** 1School of Applied Science, REVA University, Bengaluru 560064, India; 2Center for Nano and Material Sciences, Jain University, Jain Global Campus, Kanakapura, Ramanagaram, Bangalore 562112, India

**Keywords:** square wave anodic stripping voltammetry, heavy metal ions, D-valine functionalized Fe_3_O_4_

## Abstract

The functionalization of materials for ultrasensitive detection of heavy metal ions (HMIs) in the environment is crucial. Herewith, we have functionalized inexpensive and environmentally friendly Fe_3_O_4_ nanoparticles with D-valine (Fe_3_O_4_–D–Val) by a simple co-precipitation synthetic approach characterized by XRD, FE-SEM, and FTIR spectroscopy. The Fe_3_O_4_–D–Val sensor was used for the ultrasensitive detection of Cd^+2^, Pb^+2^, and Cu^+2^ in water samples. This sensor shows a very low detection limit of 11.29, 4.59, and 20.07 nM for Cd^+2^, Pb^+2^, and Cu^+2^, respectively. The detection limits are much lower than the values suggested by the world health Organization. The real water samples were also analyzed using the developed sensor.

## 1. Introduction

HMIs present in the environment are toxic and cause severe consequences. In particular, lead is very dangerous and causes a sweeping extent of physiological and behavioral dysfunction in animals and individuals [1]. Cadmium poisoning results in liver and kidney failure, and itai-itai disease [2]. Copper poising causes gastrointestinal and homeostasis disorders in humans [3]. HMIs poisoning is dangerous in long-term exposure and results in chronic illness and can be fatal. Therefore, selective and sensitive detection of HMIs is the foremost task in determining and monitoring contaminated water before supplying to the public. Among many techniques, the electrochemical detection method is highly advantageous over conventional techniques [4], such as inductively coupled plasma spectrometry [5], optical method of detection [6], atomic absorption spectroscopy [7], and surface plasma resonance spectroscopy [8]. Numerous studies have been focused to detect HMIs in water using a variety of nanomaterials, such as metal and metal oxide nanoparticles [9], polymeric nanomaterials [10], silicon [11], and carbon-based nanomaterials [12,13] to design nanosensors for the detection of HMIs. All these materials have limitations, either instability, lack of sensitivity, or difficulty with synthesis. Very recently, there has been an interest in metal oxide-based materials for sensing applications due to their ease of synthesis and higher stability than other materials [14]. Different metal oxide [15,16] electrodes are explored for electrochemical detection of HMIs [17]. For example, Wang et al. synthesized a porous graphitic carbon nitride/CoMn_2_O_4_ for the detection of HMIs, which displays a limit of detection (LOD) of 0.014 μM and 0.021 μM for Pb^+2^ and Cd^+2^, respectively [18]. Wei et al. prepared an effective electrochemical sensor by α-Fe_2_O_3_/NiO on glassy carbon (GC) electrode with a detection limit of 0.05, 0.08, 0.06, and 0.02 μM for Hg^+2^, Cd^+2^, Cu^+2^, and Pb^+2^, respectively [19]. Fan et al. synthesized ZnFe_2_O_4_ nanoparticles (ZFO) for the detection of HMIs and glucose by hydrothermal method. The reported ZFO-modified electrode exhibits excellent sensitivity, and LOD was found to be 7.38, 1.161, and 12.03 nM for Pb^+2^, Hg^+2^_,_ and Cu^+2^, respectively [20].

Among many metal oxides, iron oxide is highly suitable for electrochemical detection of HMIs due to its eco-friendliness, minimal toxicity, and low cost, along with biocompatibility. Out of numerous techniques to synthesize iron oxides, the co-precipitation method provides scalability, and hence, easy commercialization [21]. Pure iron oxide materials have a strong adsorption affinity toward HMIs [22]. Functionalization employing organic compounds [23,24,25] enhances the adsorption potential as more active moieties can be accommodated, which helps in the uptake of HMIs. In addition, functionalization prevents agglomeration and flocculation in iron oxide [26]. Functionalization of nanomaterials also aids in the sensitive detection of HMIs. It helps in chelating the HMIs from water and thus improves the accessibility to the electrode surface. Hence, very low detection limits could be achieved [27] (Figure 1).

The square wave anodic stripping voltammetry (SWASV) is highly beneficial for acquiring the peak current at high sweep rates with better sensitivity. SWASV shrinks the background noise compared to other voltametric methods and hence lower detection limits can be achieved [28]. Typically, sensing of HMIs by the SWASV method involves the adsorption of ions by electroreduction onto the surface followed by stripping into the electrolyte. The subsequent stripping current is being estimated which corresponds to the concentration of the HMIs present in the solution. To enhance the adsorption efficiency onto the electrodes, functional groups would help in chelating the metal ions from the solution which may aid in the maximum number of ions reduced on the surface of the electrode. During the process of functionalization, D-Valine was adsorbed on the surface of Fe_3_O_4_ nanoparticles. The high sensing ability of the sensor for HMIs could be due to the availability of free NH_2_ groups which chelates metal ions, and hence the adsorption and reduction process would be facilitated.

It persists that the modification of nanoparticles with small or little big organic molecules (molecules with -OH, -SH, or -NH units) will move through two different recognized mechanisms for the detection of HIMs, namely, cavity entrapment and chemical affinity (or both simultaneously) [29].

We evinced a simple, single-step pattern for easy detection of HIMs. Our approach holds the advantage of affinity between Fe_3_O_4_–D–Val (having free NH_2_ groups) and HMIs. The interface between the NH_2_ group and HIMs might follow the mechanism as shown in Figure 2a. The Fe_3_O_4_–D–Val sensor was employed for the ultrasensitive and simultaneous detection of Cd^+2^, Pb^+2^, and Cu^+2^. These nanoparticles accomplished a very low LOD of 4.59, 11.29, and 20.07 nM for Pb^+2^, Cd^+2^, and Cu^+2^, respectively. The limit of detection is considerably less than the reported values and the recommended world health organization standards. The real water samples were investigated by employing a developed sensor.

## 2. Materials and Methods

### 2.1. Instrumentation

All electrochemical measurements were performed by EG&G potentiostat/galvanostat (Model 263A) in a standard three-electrode system, saturated calomel electrode (SCE) as a reference electrode, GC as a working electrode, and Pt wire as a counter electrode. The synthesized Fe_3_O_4_–D–Val was analyzed by using an infrared spectrum recorded using FTIR (Bruker, ALPHA, Billerica, MA, USA). The XRD (Rigaku X-ray diffraction Ultima-IV) was performed to study the crystal structure of synthesized material. FE-SEM (JEOL model-JSM7100F) was carried out for the morphology of Fe_3_O_4_–D–Val.

### 2.2. Chemicals and Materials

Ferrous sulfate (crystalline) and ferric chloride anhydrous were purchased from S. D. Fine-Chem Limited (Maharashtra, India). D-valine (99+%) was purchased from Chem-Impex international (Wood Dale, IL, USA). Liquor ammonia (About 25% NH_3_) was purchased from Fisher scientific (Waltham, MA, USA). Sodium acetate buffer solution (pH 5.2 ± 0.1) was acquired from Sigma Aldrich (St. Louis, MO, USA). Cadmium nitrate tetrahydrate was obtained from LOBAchemie (Maharashtra, India). Lead nitrate was purchased from S. D. Fine-Chem Limited. Cupric sulphate pentahydrate was acquired from S. D. Fine-Chem Limited. All chemicals were utilized without any further purification. Deionized water was used for all experiments.

### 2.3. Synthesis of Functionalized Fe_3_O_4_

Fe_3_O_4_–D–Val was synthesized by the co-precipitation method as shown in Figure 2b. Typically, 2.1 g of FeSO_4._7H_2_O and 3.1 g of FeCl_3_ were dissolved in 100 mL of deionized water, and the solution was heated at 60 °C (reaction mixture). This was followed by the preparation of two solutions, i.e., (I) 10 mL of 25% liquor ammonia was dissolved in 50 mL of deionized water, (II) 0.5 g of D-valine was dissolved in 50 mL of deionized water. These two solutions were added quickly and sequentially into the solution containing the reaction mixture. The reaction mixture was kept at constant stirring with constant heating at 60 °C for about 1 h. The bright brown precipitate formed was collected after filtering followed by washing with deionized water and dried at 60 °C overnight.

### 2.4. Electrode Modification

Before electrode modification, the glassy carbon electrode (GCE) was cleaned with alumina slurry of various particle sizes (1, 0.5, and 0.25 μm average particle size) and subsequently washed with deionized water. The electrode was sonicated in water and ethanol (1:1) solution for about 30 min and washed with ethanol and dried at room temperature [30]. In the sensing system, GCE, with advantages of high cleanliness, high conductivity, minimal thermal extinction coefficient and wide potential window of operation, has been broadly utilized. In addition, the modified GC electrode provides phenomenal chemical property of immobilizing materials onto its surface, and aids insensitive and selective determination of HMIs. A total of 1 mg of the synthesized Fe_3_O_4_–D–Val was dispersed in a 1 mL mixture of water and ethanol solution (7:3) and ultra-sonicated for 30 min. A total of 10 μL of the above solution was drop-costed onto the surface of the cleaned GC and dried at room temperature.

## 3. Results and Discussion

Figure 3a shows FTIR spectra of Fe_3_O_4_–D–Val nanoparticles. The IR band at 549 cm^−1^ was attributed to the Fe-O stretching mode. Free amine group was confirmed by vibrational bands at 881 cm^−1^ attributed to the N-H wagging [31], C-C-N stretching at 1136 cm^−1^ and N-H bending at 1587 cm^−1^. The vibrational bands at 3110 and 2982 cm^−1^ correspond to the presence of O-H and C-H asymmetric stretching [32], respectively, and peak broadening confirms the presence of hydrogen (functional groups such as -OH, -CH_3_) in the synthesized nanoparticles. In the FTIR spectrum of D-valine functionalized Fe_3_O_4_ band at 1511 cm^−1^ is attributed to the N-H symmetric deformation [33].

The XRD technique (Figure 3b) was utilized to ascertain the crystalline structure of Fe_3_O_4_–D–Val. The XRD peaks with 2θ at 30.2°, 35.7°, 43.4°, 54.1°, 57.5°, 62.8°, and 71.9° exhibited consistency with reported data [34,35] (COD 96-900-2320) and were indexed to a cubic phase of Fe_3_O_4_ (magnetite, chemical formula: Fe_24.00_O_32.00_ Space group: Fd-3m). The typical crystallite size was found to be 8.62 nm.

The surface morphology of the synthesized nanoparticles is analyzed by FE-SEM; the composites consist of uniform particle distribution, and the bright nanoparticles arecovered by a carbon chain-like structure of amino acids, which can be observed in Figure 3c.

### 3.1. Electrochemical Detection of HMIs

SWASV was employed to study the electrochemical sensing of HMIs with Fe_3_O_4_–D–Val in sodium acetate buffer (0.1 M, pH 5.2 ± 0.1). Optimization of deposition potential, pH of electrolyte, and deposition time were carried out. After the optimization of experimental parameters, individual metal ion detection was carried out. Later, simultaneous detection of HIMs at optimized conditions was performed to analyze the behavior of HMIs in the presence of multiple electroactive analytes.

### 3.2. Influence of Deposition Potential

The anodic stripping analysis reveals that the suitable deposition potential appears to be key to achieving the highest sensitivity. The stimulus of deposition potential on the responsive stripping peak current of Pb^+2^ was studied by using SWASV at different potential ranges from −0.8 to −1.2 V vs. SCE in 3 μM concentration of Pb^+2^ for about 300 s of deposition time in sodium acetate buffer electrolyte (0.1 M and 5.2 pH) as presented in Figure 4a,b. The peak current increases with increasing deposition potential from −0.8 to −1.1 V. Deposition potential beyond −1.1 V resulted in the current decreasing. The decrease in the peak current beyond −1.1 V may be partially due to the hydrogen evolution observed at the surface of the electrode. The adsorption of the hydrogen ion intermediates on the surface of the electrode results in fewer active sites for adsorption and electro–reduction in the HMIs, thereby resulting in a reduced peak current [36].

### 3.3. Influence of pH of the Electrolyte

Figure 4c,d, represents the peak current of sensing Pb^+2^ and its relationship with pH; the increase in the pH results in the peak current increasing up to 5.2 pH and further expansion in pH prompts a decline in the peak current. At very low pH, protons compete with HMIs to bind to the electrode surface. As pH increases, the availability of protons decreases, and hence adsorption HMIs increases. At higher pH, HMIs undergo hydrolysis, which results in the peak current decreasing [37,38].

### 3.4. Influence of Deposition Time

The effect of the deposition time on the sensing ability was studied (Figure 4e,f); as depicted there is an increase in the peak current when the deposition current is increased from 60 to 300 s with an increment of 60 s in each successive step. The optimized deposition time as observed from the figure is 300 s and this parameter is used for further analysis irrespective of the HMIs.

### 3.5. Individual Detection of HMIs (Pb^+2^, Cd^+2^, and Cu^+2^) by SWASV

To detect the HMIs, SWASV was applied to study the GC-modified Fe_3_O_4_–D–Val as HMIs sensor in 0.1 M sodium acetate buffer electrolyte under optimized conditions. The detection of Pb^+2^ was studied by increasing the concentration of Pb^+2^ from 0.08 to 2 μM as shown in Figure 5a,b. Linear increase in anodic peak current response with increasing concentration of Pb^+2^ is obvious. The oxidation peak for detection of Pb^+2^ was at −0.51 V vs. SCE. We have observed two linear concentration ranges. At low concentration, Figure 5a and high concentration Figure 5b show linear equation Y = 0.090 X + 2.099 and Y = 0.019 X + 20.458, respectively. The LOD and sensitivity were calculated using the formula LOD = 3S/b (where S corresponds to standard deviation and b corresponds to slope from the calibration curve) and sensitivity = b/A (where A represents the area of GCE).

In addition, SWASV was applied to examine the detection of other metal ions such as cadmium (Cd^+2^) and copper (Cu^+2^), at concentrations from 0.05 to 0.8 μM and 0.1 to 1 μM, respectively. The oxidation peaks for the detection of Cd^+2^ and Cu^+2^ was observed at −0.67 and −0.07 V vs. SCE, respectively. The corresponding calibration plot was plotted as shown in Figure 5c,d. The linear equations for Cd^+2^ and Cu^+2^ are Y = 0.036 X − 0.728 and Y = 0.020 X + 6.728, respectively.

### 3.6. Simultaneous Detection of the HMIs by SWASV

Simultaneous detection was conducted by increasing the concentration of Cd^+2^, Pb^+2^, and Cu^+2^. Oxidative stripping peak was observed at −0.67 V, −0.51 V, and −0.007 V vs. SCE for Cd^+2^, Pb^+2^, and Cu^+2^, respectively, as displayed in Figure 6a,b. The oxidative stripping peak current increases with increasing concentration. The interpeak spacing is adequate to distinguish between the different metal ions. The anodic spacing between Cd^+2^ and Pb^+2^ is about −0.1675 V vs. SCE and −0.4469 V vs. SCE between Pb^+2^ and Cu^+2^. The LODs for Pb^+2^, Cd^+2^, and Cu^+2^ by simultaneous detection are 18.89, 18.38, and 7.48 nM, respectively. Comparison of different electrode material LODs are listed in Table 1. The LODs for simultaneous and individual HIMs detection of as synthesized electrode material with their thresh hold limits by WHO is shown in Table 2. Compared to individual HMIs detection, the results appear to rise in LOD values (Pb^+2^ by 14.33 nM and Cd^+2^ by 7.09 nM). Cu^+2^ showed comparatively low LOD (Cu^+2^ by 12.609 nM), which could be ascribed to the interface between different HMIs, such as the development of intermetallic compounds and competitive deposition [39]. The outcomes from simultaneous detection point to the application of developed sensors for real-time water analysis.

### 3.7. Reproducibility, Stability and Interference Study’s

Fe_3_O_4_–D–Val/GC-modified electrode exhibits electrochemical detection towards Cd^+2^, Pb^+2^, and Cu^+2^. To evaluate the reproducibility of the sensor, SWASV was employed to detect 1 μM Pb^+2^, 2 μM Cd^+2^, and 2 μM Cu^+2^ in sodium acetate buffer (0.1 M) for ten separate trials. The relative standard deviation (RSD) is 2.37%, 7.94% and 11.54%, for Pb^+2^, Cd^+2^ and Cu^+2^ ions, respectively. The stability of the electrode was also analyzed under similar conditions, and the anodic stripping current response remains at 78.04%, 85.10%, and 73.66% even after 16 days as shown in the Figure 6c,d. Further, to evaluate the interference of other metal ions on the as-prepared sensor, 1 μM Pb^2+^ was followed by 0.1 μM of conceivable interfering metal ions such as Cd^2+^, Fe^2+^, Cr^3+^, Hg^2+^, Ni^2+^, and Cu^2+^. It was found that there is no prevalent change in ht epeak current of Pb^2+^ ((M) = 1 μM) as depicted in Figure 6f.

### 3.8. Real-Time Applications Study

As per the world health organization, the permissible concentration of Pb^2+^ in drinking or river water is 10 μg/L. As depicted in Figure 5a,b, the Fe_3_O_4_–D–Val electrode allows for detection as low as WHO permissible limits. On this basis, Fe_3_O_4_–D–Val electrodes were employed for the analysis of real water samples (River water: 13°45′11.6″N 76°53′45.3″, 200F and Time 2:20 PM). The real sample shows no peak corresponding to the heavy metal ion. To confirm this further, the actual sample was spiked with Pb^2+^ ions. The samples were spiked in a 3:7 volume ratio of real water sample and 0.1 M sodium acetate buffer [51]. The spiked sample displays a peak corresponding to the Pb^2+^ ions. However, the peak current in real sample water is less than in the clean water when measuring individual Pb^+2^ ions, as is evident in Figure 6e. This could be due to the influence of other natural impurities and industrial effluents in a real sample as well as a lack of ionic species in the electrolyte, thereby reducing the ionic conductivity in the electrolyte and resulting in a lesser anodic peak current.

## 4. Conclusions

An easy, cost-efficient, and sustainable synthesis pathway for the preparation of iron NPs-integrated D-Valine materials has been developed and successfully employed for SWASV detection of HMIs, i.e., Cd^+2^, Pb^+2^, and Cu^+2^. The Fe_3_O_4_–D–Val-modified GCE was found to exhibit excellent selectivity, sensitivity, and low LOD for selective as well as simultaneous detection of HMIs. The modified electrode was also found to detect HMIs in the real water sample. This positive result in this work evokes the use of Fe_3_O_4_–D–Val/GCE as a sensing platform for the detection of HMIs in the water.

## Figures and Tables

**Figure 1 sensors-22-08895-f001:**
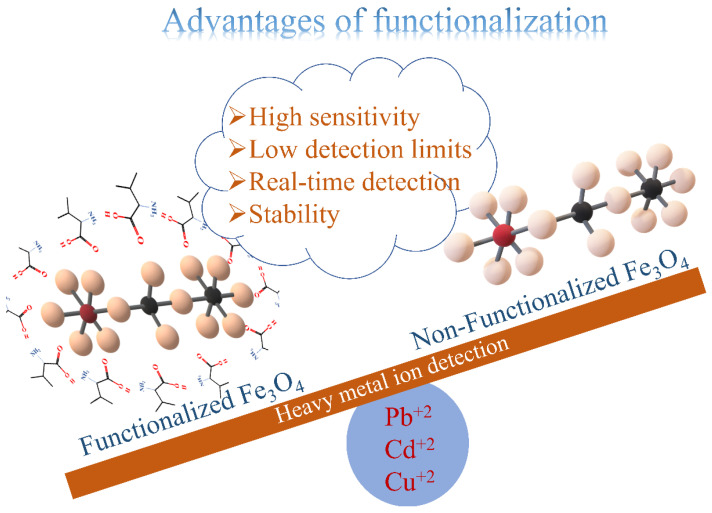
Advantages of functionalized Fe_3_O_4_.

**Figure 2 sensors-22-08895-f002:**
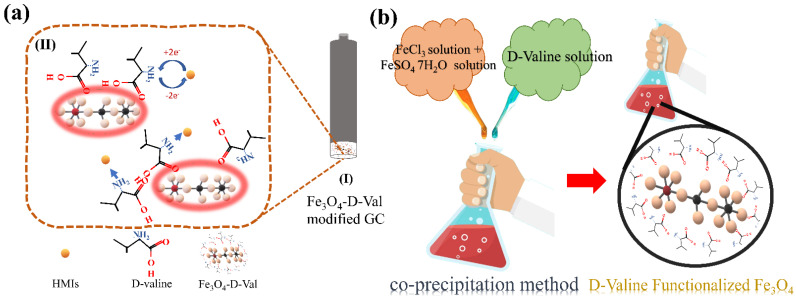
(**a**) A possible mechanism of interaction of HIMs with free NH_2_ groups of Fe_3_O_4_–D–Val, (**I**) Fe_3_O_4_–D–Val modified glassy carbon electrode, and (**II**) Interaction of HMIs with NH_2_ groups of Fe_3_O_4_–D–Val. (**b**) Schematic synthesis of Fe_3_O_4_–D–Val by co-precipitation process.

**Figure 3 sensors-22-08895-f003:**
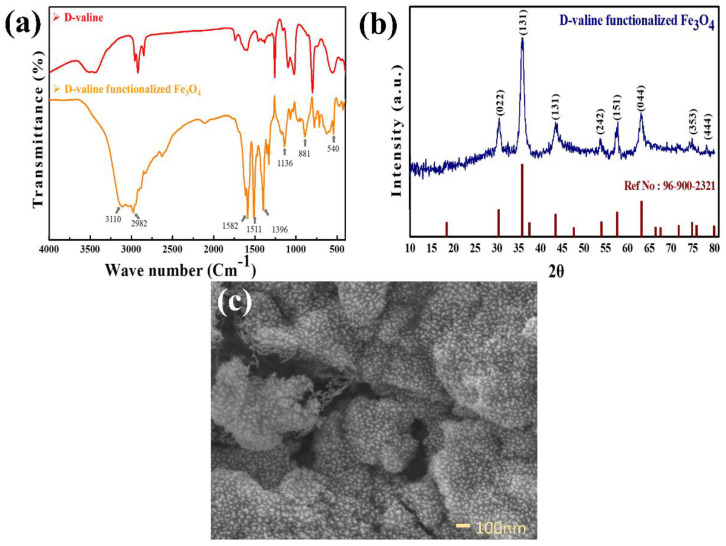
(**a**) FT-IR spectra of Fe_3_O_4_–D–Val. (**b**) XRD pattern of Fe_3_O_4_–D–Val. (**c**) FE-SEM images of Fe_3_O_4_–D–Val.

**Figure 4 sensors-22-08895-f004:**
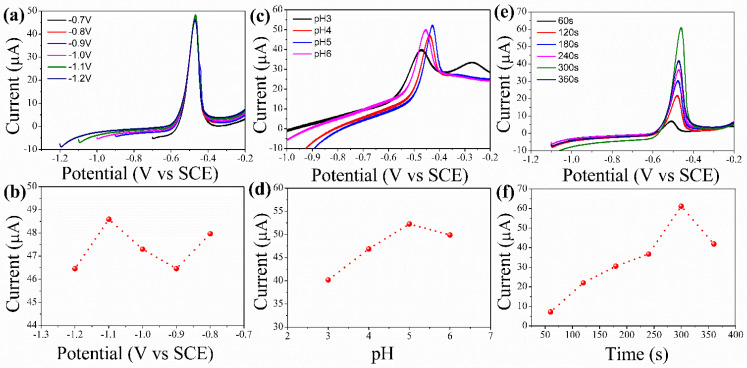
(**a**) voltammogram of optimization of deposition potential, and (**b**) Optimization of deposition potential by applying SWASV at different potentials (300 s deposition time and (Pb^+2^) = 3 μM). (**c**) voltammogram of optimization of pH of the electrolyte, and (**d**) Optimization of pH of the electrolyte (optimized deposition potential: −1.1 V vs. SCE, deposition time: 300 s and (Pb^+2^) = 3 μM). (**e**) voltammogram of optimization of deposition time, and (**f**) Optimization of deposition time by SWASV (0.1 M sodium acetate buffer electrolyte, deposition potential −1.1 V vs. SCE and (Pb^+2^) = 3 μM).

**Figure 5 sensors-22-08895-f005:**
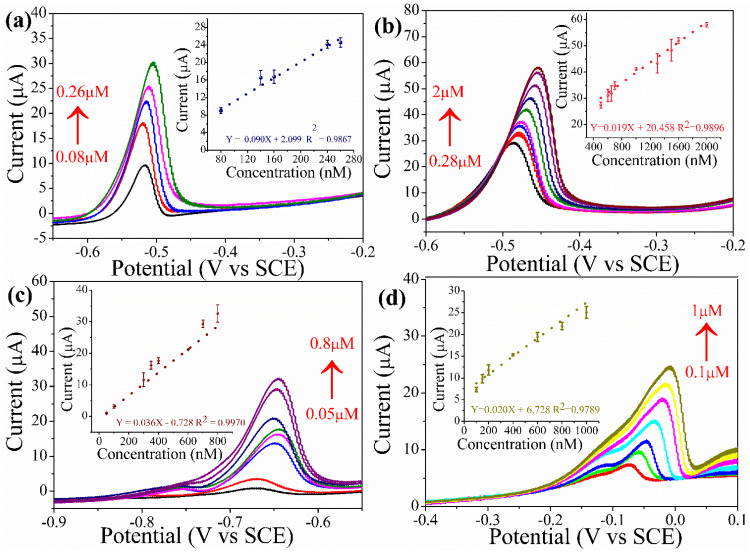
(**a**) SWASV of Pb^+2^ was applied on modified glassy carbon under optimized conditions at low and (**b**) high concentrations. (**c**) SWASV detection of Cd^+2^ and (**d**) SWASV detection of Cu^+2^ under optimized conditions. Different color line’s corresponds to the different concentration of heavy metal ions. With in graph its difficult to display all concentartions, the concentartion rage from minimal to maximum is displayed with upword arrow mark.

**Figure 6 sensors-22-08895-f006:**
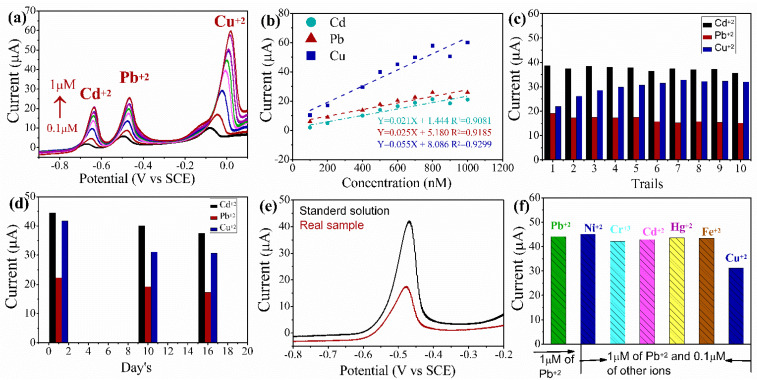
(**a**) SWASV for simultaneous detection of three different metal ions at optimized conditions and (**b**) Calibration plot from simultaneous detection. (**c**) Anodic stripping response by SWASV for reproducibility of Fe_3_O_4_–D–Val_/_GC-modified electrode for detection of 1 μM Pb^+2^, 2 μM Cd^+2^, and 2 μM Cu^+2^ in 0.1 M sodium acetate buffer solution. (**d**) Stability of the modified glassy carbon electrode at a constant concentration of HMIs (2 μM). (**e**) SWASV peak spiking of Pb^+2^ at 1 μM concentration. (**f**) Effect of interference species. Different color line’s corresponds to the different concentration of heavy metal ions. With in graph its difficult to display all concentartions, the concentartion rage from minimal to maximum is displayed with upword arrow mark.

**Table 1 sensors-22-08895-t001:** Comparison of LOD and sensitivity of different electrodes reported in the literature.

Modification Material	Measurement Technique	LOD			Sensitivity	Ref
		Pb^+2^	Cd^+2^	Cu^+2^		
Pg-C_3_N_4_/CoMn_2_O_4_	SWASV	0.014 μM	0.021 μM		Pb^+2^:22.39 (μA/μM cm^2^)	[18]
CI-DPTU/GCE	SWASV	11.0 nM	6.45 nM	7.85 nM		[40]
ae-Fe/Fe_2_O_3_@cc	DPASV	0.5 ppb	0.42 ppb		Pb^+2^: 408.0 Cd^+2^: 338.7 (μA/μM cm^2^)	[41]
Co_3_O_4_-NC/SPCE	DPV	0.00722 μM		0.00173 μM	Pb^+2^: 16.73 Cu^+2^: 11.46 (μA/μM cm^2^)	[42]
UiO-66- NH_2_/GaOOH	DPV	0.028 μM	0.016 μM	0.019 μM		[43]
UiO-66/Bi/GCE	SWASV	0.94 µg/L	2.01 µg/L			[44]
Fe-OSA	DPV	0.0360 μM	0.0192 μM			[45]
SBDDE	SWASV	5–120 μg/L			Pb^+2^: 0.42 (μA/μM cm^2^)	[46]
S-doped C_3_N_4_ tube bundles/graphene nanosheets composite	SWASV	0.78 nM	2.30 nM			[47]
Pd1.5/PAC-900	DPV	50 nM	41 nM	66 nM	Pb^+2^: 109.1 Cd^+2^: 72.9 Cu^+2^: 21.8 (μA/μM cm^2^)	[48]
Mg–Al-LDH/Nafion	SWASV		0.20 nM		Cd^+2^: 13.86 mA mM^−1^	[49]
CB-15-crown-5-GEC, GC/FcIB15C5	DPV, SWASV			2.3, 0.11 g/L		[50]
D-valine functionalized Fe_3_O_4_	SWASV	4.59 nM	11.29 nM	20.07 nM	Pb^+2^: 1.275 Cd^+2^: 0.518 Cu^+2^: 0.291 (μA/nM cm^2^)	This work

**Table 2 sensors-22-08895-t002:** Comparison of detection limits obtained for Cd^+2^, Pb^+2^, and Cu^+2^ with the Fe_3_O_4_–D–Val sensor and WHO [30] threshold limits.

Metal Ions	LOD a (nM)	Sensitivity (μA/nM cm^2^)	LOD b (nM)	Sensitivity (μA/nM cm^2^)	WHO Standard Values (mg/L)
Pb^+2^	4.59	1.275	18.89	0.3101	0.01
Cd^+2^	11.29	0.518	18.38	0.3186	0.003
Cu^+2^	20.07	0.291	7.481	0.7832	2.0

a: LODs by individual ion detection, b: LODs by simultaneous detection.

## Data Availability

Not applicable.

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
