# Peer review of "Tuning the Surface Functionality of Fe3O4 for Sensitive and Selective Detection of Heavy Metal Ions"

_sensors, 2022, doi:10.3390/s22228895_

Round 1
Reviewer 1 Report
Overall: While the work is interesting, they only provide the results of the capability of sensor, and the discussion is rather poor. They need to provide data in concise and prove novelty about their work. I see a lot of spelling mistakes though out (including fig.6) from and address the following comments as this manuscript with the current form isn’t apt.
1. page 1, Introduction section, line 35 “Electrochemical methods offer portability, excellent sensitivity, short analysis time, and most importantly low cost”. The advantages are not specific to electrochemical sensors alone, thus I recommend to remove it.
2. Nowhere in the draft authors abbreviated SWASV. What is the rationale behind using this technique? Are there any advantages in GCEs or the type of detection materials (iron oxide)?
3. As shown in Fig.5, the output current has some cross-sensitivity issues. 0.8 uM Cd+2 response syncs with 0.26 uM Pb+2. How do authors justify this cross-sensitivity?
4. In Fig.6e, why is there a difference a difference, authors need to conduct more experiments on the selectivity aspect. In regular water there are minerals that do act as interferons and jeopardize the peaks in Fig. 6a.
5. What is the lifetime of this sensor? Realistically how long can it be effective under shelf storage? How does humidity and temperature effect the measurements? These experiments need to be conducted.
Reviewer 2 Report
The authors proposed an electrochemical sensor for heavy metal ions based on Fe3O4-D-Val. And The analytical performance of this sensor is well. However, the paper suffers from several problems that need to be made a minor revision in order to eliminate scientific flaws.
1. The Lattice structure of Fe3O4-D-Val should be characterized by TEM.
2. The influence of other metals ions, such as Hg2+, Cr3+, Ni2+, Zn2+, should be investigated.
3. Does the carboxyl group (C=O) be observed in FT-IR spectrum?
4. The English should be polished.
Round 2
Reviewer 1 Report
This articles is in good shape now, it could be accepted for publication.